

# Comparison of uncertainty in multi-parameter and multi-model ensemble hydrologic analysis of climate change

Younggu Her[1], Seung-Hwan Yoo[2], Chounghyun Seong[3], Jaehak Jeong[4], Jaepil Cho[5], Syewoon Hwang[6]

[1]Department of Agricultural and Biological Engineering & Tropical Research and Education Center, University of Florida, Homestead, FL 33031, United States
[2]Department of Rural and Bio-Systems Engineering, Chonnam National University, Gwangju 500-757, Republic of Korea
[3]Department of Biological Systems Engineering, Virginia Tech, Blacksburg, VA 24601, United States
[4]Texas A&M AgriLife Research, Texas A&M University, Temple, TX 76502, United States
[5]Research Department, APEC Climate Center, Busan 612-020, Republic of Korea
[6]Department of Agricultural Engineering, Gyeongsang National University, Jinju 660-701, Republic of Korea

*Correspondence to*: Seung-Hwan Yoo (yoosh15@jnu.ac.kr)

**Abstract.** Quantification of uncertainty in ensemble based predictions of climate change and the corresponding hydrologic impact is necessary for the development of robust climate change adaptation plans. Although the equifinality of hydrological modeling has been discussed for a long time, its impact on the hydrologic analysis of climate change has not been studied enough to provide clear ideas that represent the relative contributions of uncertainty contained in both multi-GCM (general circulation model) and multi-parameter ensembles toward the projections of hydrologic components. This study demonstrated that the uncertainty in multi-GCM (or multi-model) ensembles could be an order of magnitude larger than that of multi-parameter ensembles for predictions of direct runoff, suggesting that the selection of appropriate GCMs should be much more emphasized than the selection of a parameter set among behavioral ones when projecting direct runoff. When simulating soil moisture and groundwater, on the other hand, equifinality in hydrologic modeling was more influential than uncertainty in the multi-GCM ensemble. Also, uncertainty in a hydrologic simulation of climate change impact was much more closely associated with uncertainty in ensemble projections of precipitation than that in projected temperature, indicating a need to pay closer attention to the precipitation data for improvement of the reliability of hydrologic predictions. From among 35 GCMs incorporated, this study identified GCMs that contributed the most and least to uncertainty in an assessment of climate change impacts on the hydrology of 61 Ohio River watersheds, thereby exhibiting a framework to quantify contributions of individual GCMs to the overall uncertainty in climate change modeling.



# 1 Introduction

General circulation models (GCMs) have been developed by many national and international research institutions and agencies and served as useful, and probably the only, tools to predict future climate change (Murphy et al., 2004; Pierce et al., 2009; Overland et al., 2010). Since each GCM has been developed based on its own assumptions and unique mathematical representations of physical climate system processes, different climate change estimations are provided (Hawkins and Sutton, 2009). Thus, climate model selection is not only a watershed modeler's first decision in a hydrologic analysis of climate change, but it is also one of the most critical tasks though it is often undertaken with limited information regarding quality and reliability (Murphy et al., 2004). The Intergovernmental Panel on Climate Change (IPCC) launched the Coupled Model Intercomparison Project Phase 5 (CMIP5) in the fifth Assessment Report (AR5), whereby a multi-model ensemble analysis was facilitated through the provision of climate model outputs that comply with community standards (Taylor et al., 2012; IPCC, 2013; Sansom et al., 2013). The multiple general circulation model (multi-GCM) ensembles has served as a framework for accommodating probabilistic approaches in interpretation of climate change predictions and decision-making processes, and many studies have attempted to quantify uncertainty and identify its sources (Nohara et al., 2006; Christensen and Lettenmaier, 2007; Tebaldi and Knutti, 2007; Graham et al., 2007; Sheshukov et al., 2011; Chong-Hai and Ying, 2012; Harding et al., 2012; Velázquez et al., 2013). Ensemble averaging can improve the accuracy of a climate projection by allowing GCM errors cancel each other out (Pierce et al., 2009). However, the approach often does not employ all models available thus may underestimate uncertainty and/or produce a bias in the ensemble prediction (Tebaldi and Knutti, 2007; Vander Linden and Mitchell, 2009). Further, interpretation of an ensemble averaging prediction remains challenging due to "the lack of consensus on combing models" (Knutti et al., 2010; Parker, 2010).

Because of the global nature of the climate system and the complexity of the underlying climate physics, climate change impact assessments are often implemented in continental and regional extents, which, however, are not the scales at which most hydrologic analyses and water resources managements are carried out (Hostetler, 1994; Xu, 1999a; 1999b; Arora and Boer, 2001; Stone et al., 2001; Guo et al., 2002; Varis et al., 2004; Nohara et al. 2006; Döll and Schmied, 2012; Oubeidillah et al., 2013). A large-scale analysis may not consider detailed hydrological processes, and localized impacts may not be effectively represented at such scale (Varis et al., 2004; Hulme, 2005; Young et al., 2009). For instance, hillslope processes including infiltration and overland flow transport are more dominant and influential in hydrology and ecosystem of a small watershed, while channel routing and groundwater flow are more important processes controlling the overall hydrologic response of a large watershed (Huff et al., 1982; Ward, 1984; Meyer et al., 2007; Richardson and Danehy, 2007; Frisbee et al., 2011). In addition, it is reasonable to assume a homogeneous landscape for a hillslope, whereas a large-scale watershed tends to have great heterogeneity in its landscape (Hostetler, 1994). The responses of individual landscape units of a large watershed are likely to be intermingled with others and dampened through prolonged overland and channel processes



(Frisbee et al., 2011; Stanfield and Jackson, 2011; Frisbee et al., 2012). The hydrological responses of local head watersheds to climate change would, therefore, be clearly explained at small spatial scales.

Many different hydrological models, from distributed to lumped, have been utilized in climate change studies: the variable infiltration capacity (VIC) model (Christensen and Lettenmaier, 2007; Hayhoe et al. 2007; Oubeidillah et al., 2013), Hydrologiska Byråns Vattenbalansavdelning (HBV) model (Bergström et al. 2001; Arheimer et al., 2005; Graham et al., 2007; Akhtar et al., 2009), and the Soil and Water Assessment Tool (SWAT) model (Stone and Hotchkiss, 2003; Graiprab et al., 2010; Mango et al., 2011; Sheshukov et al., 2011; Van Liew et al., 2013), as well as simple models such as ABCD and Budyko (Fu et al., 2007, Sankarasubramanian and Vogel, 2002; Tigkas et al., 2012; Liu and Cui, 2011; Liu and Yang, 2010; Dooge, 1992; Poff et al., 1996). Complicated models can simulate detailed hydrologic processes, but the sizable input data and parameter requirements tend to result in uncertainty (Her and Chaubey, 2015). Simpler models are therefore preferable as long as they can provide predictions regarding hydrological variables and components of interest at the required levels of accuracy and detail, especially when the overall far future hydrologic responses of a watershed are of interest.

An understanding of the sources and influences of uncertainty helps to identify the ways that can efficiently improve the robustness and reliability of a climate change impact analysis, whereby the subsequent development of climate change mitigation strategies and water resource management plans can be more effective. Equifinality is one of the main sources of uncertainty in hydrologic modeling, and many methods have been proposed to quantify equifinality and the resulting uncertainty (Beven, 2006; Sadegh and Vrugt, 2013). While it is known that equifinality decreases with an increase in the number of observations and a decrease in the number of calibration parameters, equifinality is inevitable, and its impact is substantial in hydrologic modeling (Her and Chaubey, 2015). There are only a few known studies about the influence of equifinality of hydrological models on climate change impact assessment. Poulin et al. (2011) demonstrated that hydrologic model structure uncertainty is more influential than parameter uncertainty on assessment of climate change impact in a snow dominated river basin. Maurer et al. (2010) found that a climate change impact assessment could be significantly affected by hydrologic model selection and parameter calibration. Several studies showed that selection of hydrologic model (structural uncertainty) is much more influential than GCM selection in assessments of climate change (Kay et al., 2009; Gosling et al., 2011; Najafi et al., 2011). Chen et al. (2010) and Dobler et al. (2012) demonstrated that, in terms of a climate change impact assessment, hydrologic model parameter uncertainty is the least influential; notably, though, the numbers of the arbitrarily selected unique parameter sets incorporated for their studies were only 10 and 20, respectively, indicating high possibility of underestimation in the assessments of the equifinality impacts. In addition, most hydrological analyses of climate change uncertainty employed a single or a few study watersheds and/or the phase 3 of the Coupled Model Intercomparison Project (CMIP3) in their case studies even though it has been several years since the latest climate models of CMIP5 built under RCP scenarios were first released, meaning that the applicability of the study results is possibly limited.

This study compared the significance of selections of GCMs and hydrological model parameters (equifinality) on hydrological assessment of climate change by quantifying uncertainty in multi-GCM and multi-parameter ensemble





projections for weather and hydrology of multiple watersheds selected within the Ohio River basin. In this study, 35 ensemble members including 22 CMIP5 GCMs and their variants (hereafter 35 GCMs) were considered, and 61 study watersheds were incorporated to show the variability of the quantified uncertainty across the different watersheds. A simple, monthly water balance model, ABCD, was employed as a mathematical representation of the mechanisms that control the

5    responses of the hydrologic components to climate variability. The behavioral parameter sets of the water balance model that were developed for each watershed were identified using the Generalized Likelihood Uncertainty Estimator (GLUE) framework (Beven and Freer, 2001), and multiple thresholds were applied to see the sensitivity of the equifinality contributions to subjectivity.

## 10    2 Methods and Materials

### 2.1 Study area

The Ohio River basin is located in the Eastern Corn Belt and extends across nine states from Illinois to New York, between the latitudes 36° 07' and 42° 26' North, and the longitudes 77° 50' and 89° 01' West. The basin drains a primarily agricultural area of 374,000 km$^2$ including several large cities into the Mississippi River and eventually the Gulf of Mexico.

For this study, 61 study watersheds within the Ohio River basin were selected for the consideration of the drainage areas, locations, availability of streamflow measurements, and applicability of the ABCD model (Figure 1). The total drainage area of the selected watersheds is 41,341 km$^2$, (average size is 678 km$^2$) which is 11% of the entire basin's drainage area. The daily precipitation and temperature observations that were obtained from 103 weather stations associated with the basin were used for statistical downscaling of GCM climate projections. The climate of the Ohio basin varies from humid subtropical

(north-east) to humid continental (southwest), and the annual average temperature and precipitation across the basin are 11.3° C and 1,032 mm, respectively.

Chien et al. (2013) predicted that the annual streamflow of the agricultural watersheds in the Midwestern United States would decrease by as much as 40% under the Speical Report on Emission Scenarios (SRES) of 26 GCM projections. Panagopoulos et al. (2015) found that the crop productivity of the Ohio River basin could decrease by 20% under climate

projections; they also found large amounts of uncertainty in the sediment and nutrient loads of the basin that were projected by seven CMIP5 GCMs. Ebner et al. (2015) investigated the impacts of future climate changes, represented by four CMIP3 GCMs, on the hydrology of the Upper Scioto River Basin that drains 8,337 km$^2$ into the Ohio Basin; depending on the GCMs used in their study, the annual streamflow projections varied by a factor of two to three, indicating a large uncertainty in the multi-GCM ensembles for the study basin. Kunke et al. (2013) investigated the future climate scenarios of the

Midwest U.S. projected by 15 CMIP3 GCMs, and found wide variations of the annual precipitation and temperature projections, depending on the GCMs that are employed.



## 2.2 Multi-GCM ensemble

Over the last several years, the climate projections from the GCMs participating in the CMIP3 and CMIP5 have been employed for climate change impact assessments at both regional and local scales (Lopez et. al, 2009; Johnson et. al., 2011). CMIP5, the latest climate data, is expected to promote multi-GCM frameworks by providing a range of projected climate sciences (Taylor et al., 2012). In this study, climate projections for the weather gage stations associated with the study watersheds were obtained by downscaling the 35 climate change outputs selected from 22 GCMs of the CMIP5 (Figure 1 and Table 1). In addition, two RCP scenarios (RCP 4.5 and RCP 8.5) that have been commonly adopted as forcing scenarios for the CMIP5 GCMs were employed to consider uncertainty in the future social conditions (van Vuuren et al., 2011), leading to the formulation of 35 climate projections for each of the combinations of the RCP scenarios and watersheds in this study (Table 1).

Because of the inconsistency regarding the spatial resolutions between GCM data and a climate change impact assessment, GCM data are often downscaled to finer resolutions, and often into existing weather stations (Christensen et al., 2008; Maraun, 2013). For this study, the CMIP5 GCM outputs (precipitation, maximum and minimum temperatures) of the Ohio River study watersheds were statistically downscaled over the period from 1950 to 2099 using the hybrid semi-parametric approach proposed by Ho et al., (2012), which is considered computationally efficient and easy to implement (Diaz-Nieto and Wilby, 2005). Using Eq. (1), the approach matches the location (mean), scale (variance), and shape (skewness) parameters of the climate change data with those of the historical data to preserve the consistency of the data's statistical features over long term periods:

$$\hat{X}'_o = \mu_o + \frac{\sigma_o}{\sigma_\mu}(X'_m - \mu_m),\tag{1}$$

where X, μ, and σ respectively represent the variable of interest, mean, and standard deviation of a climate, the subscripts of o and m respectively signify the observable and simulated climate variables of interest, the superscript of " ' " indicates a future period, and the symbol of "^" represents a bias corrected variable. Once the GCM downscaling into the existing weather stations was completed, the multi-GCM ensemble averages of the weather variables (precipitation and temperature) and the hydrologic components were determined by averaging the downscaled projections with equal weights according to the "one model, one vote" weighting scheme (Sansom et al., 2013).

## 2.3 Hydrologic model

A simple hydrologic model, ABCD was prepared to simulate the long-term monthly hydrologic responses of the 61 study watersheds to projected climate changes (Thomas, 1981). In the ABCD model, available water ($WW$, mm) of the current month is defined as a summation of precipitation ($PP$, mm) of the current month ($t$) and soil water content ($SS$, mm) of the previous month ($t-1$) (Eq. (2)), while the evapotranspiration opportunity of the current month ($YY$, mm) is determined by a summation of actual evapotranspiration and soil water content of the current month (Eq. (3)), as follows:

$$WW_t = PP_t + SS_{t-1}\tag{2}$$



and

$$YY_t = PET_t + SS_t = \frac{WW_t+b}{2a} - \sqrt{\left(\frac{WW_t+b}{2a}\right)^2 - \frac{WW_tb}{2a}} \quad , \tag{3}$$

where the $a$ and $b$ parameters represent "propensity for runoff to occur well before the soil is saturated to capacity" ($0 \leq a \leq 1$) and "upper limit of storage in the unsaturated zone above the groundwater level," or "upper bound of the summation

of actual evapotranspiration and soil moisture storage," respectively (Thomas, 1981). $PET_t$ represents the potential evapotranspiration (mm) that is calculated using an equation such as the following Penman and Hargreaves equation (Eq. (4)):

$$PET = e \cdot PET_{EQ}, \tag{4}$$

where $e$ is a calibration parameter that is newly introduced to the original ABCD model, and $PET_{EQ}$ is the potential

evapotranspiration estimation provided by the $PET$ equation. Due to its simplicity, the following Hargreaves equation was selected for a calculation of the monthly $PET$ in this study (Eq. (5)):

$$PET_{EQ} = 0.000938(TAV + 17.8)(TMX - TMN)^{0.5}R_a, \tag{5}$$

where $TAV$ is average monthly temperature ($^\circ$ C ), $TMX$ is maximum monthly temperature ($^\circ$C), $TMN$ is minimum monthly temperature ($^\circ$C), and $R_a$ is extraterrestrial radiation ($MJm^{-2}month^{-1}$).

In the ABCD model, the soil water content is proportional to the evapotranspiration opportunity, and it exponentially increases with increases of the potential evapotranspiration rate (Eq. (6)), as follows:

$$SS_t = YY_t exp\left(\frac{-PET_t}{b}\right) \tag{6}$$

Groundwater storage ($GG$, mm) and streamflow (or total runoff: $QQ$, mm) are calculated as functions of the available water and the evapotranspiration opportunity using Eq. (7) and Eq. (8), respectively, as follows:

$$GG_t = GG_{t-1} + c(WW_t - YY_t) - dGG_t \tag{7}$$

and

$$QQ_t = (1 - c)(WW_t - YY_t) + dGG_t, \tag{8}$$

where $c$ is a parameter that is equivalent to the baseflow index and represents a fraction of the streamflow contributed by the groundwater, and $d$ is the groundwater residence time that is proportional to the baseflow recession constant.

Evapotranspiration ($ET$, mm) is then regarded as the difference between the precipitation and the total runoff. Further, the middle term on the right side of Equation 7 is  groundwater recharge, and the left- and right-side terms on the right side of Equation 8 represent direct runoff ($DR$, mm) and groundwater discharge ($GW$, mm), respectively.

## 2.4 Multi-parameter ensemble

The ABCD model prepared for each watershed was calibrated to the monthly streamflow measured at the watershed outlet.

A sampling based optimization algorithm, Shuffled Complex Evolution – University Arizona (SCE-UA) (Duan et al., 1992; 1994), was used to explore the parameter space and to find sets of the five parameters, $a, b, c, d$, and $k_c$ that provide





acceptable model performance statistics during the calibration period from 1990 to 2012. In the calibration, multiple parameter sets that satisfy the predefined performance requirements were identified as behavioral sets under the GLUE framework (Beven and Freer, 2001). These behavioral sets are defined as "equally good" and "equally acceptable" (Her and Chaubey, 2015). To take subjectivity into account in the parameter uncertainty estimation, the combinations of an absolute

threshold of the minimum NSE of 0.67 and the four different relative thresholds of the best 10%, 7.5%, 5%, and 2.5% were applied to identify the behavioral parameter sets out of those sampled in the calibration. It is worth noting that this study initially included 156 candidate watersheds from within the Ohio River basin for which USGS streamflow gage data are available, and those watersheds with ABCD models that did not meet the absolute performance criterion (NSE of at least 0.67) were not included in this study.

**2.5 Quantification of uncertainty in multi-parameter and multi-GCM ensembles**

A range of the difference between the maximum and minimum values was used as a measure of the amounts of uncertainty in the ensemble predictions that were made using multiple GCMs and behavioral parameter sets. For this study, uncertainty in the multi-parameter ensembles was first quantified by calculating the ranges (upper limits minus lower limits) of the monthly hydrographs simulated using the behavioral parameter sets that had been previously identified for each combination

of GCMs and study watersheds (Figure 2). Then, an average hydrograph of multi-parameter ensembles was derived for each GCM and study watershed combination, and the range of the variations in the average hydrographs across the GCMs for each watershed was regarded as the amounts of uncertainty in the multi-GCM ensembles (Figure 2). Thus, the uncertainty amounts quantified for two difference sources, a multi-parameter ensemble and a multi-GCM ensemble, became independent of each other, which allows direct comparison of the two uncertainty quantities.

The contribution of each GCM model to the GCM model selection uncertainty was quantified by comparing the uncertainty amounts (ranges) in either the GCM ensemble predictions of the monthly climate variables or the hydrologic components that are made with/without the use of each GCM (Equation (9)), as follows:

$$U^Q(GCM_x) = U^Q(GCM_{\forall x \in S}) - U^Q(GCM_{x \notin S}),$$     (9)

where $U^Q(GCM_x)$ is the uncertainty quantities in the GCM ensemble predictions for either a climate variable or a hydrologic

component $Q$ (e.g., PP and QQ), which are solely attributed to $GCM_x$; $U^Q(GCM_{\forall x \in S})$ is the total uncertainty in the entire GCM ensemble; and $U^Q(GCM_{x \notin S})$ is the measured uncertainty in the GCM ensemble for which $GCM_x$ is excluded. The relationships between the uncertainty quantities in the ensemble projections of the climate variables and those in the ensemble projections of the hydrologic components were then investigated to see which climate variable (precipitation, maximum and minimum temperatures) exerts the most significant influence on the hydrologic prediction uncertainty.





## 3 Results and Discussion

### 3.1 Projected precipitation and temperature

The precipitation projections made by the 35 GCMs were averaged by the months and by the study watersheds to investigate the overall trends of future precipitation in the Ohio River basin. The annual average precipitation of the Ohio River watersheds from 2020 to 2099 was projected to increase by 6.8 % and 8.8 % under the RCP 4.5 and RCP 8.5 scenarios, respectively. The projected monthly precipitations showed large seasonal variations, with up to 14% and 19% increases under the RCP 4.5 and RCP 8.5 scenarios, respectively (Figures 3 (a) and (b)). The increase rates were higher in winter and spring than in summer, which is in agreement with the findings of Kunkel et al. (2013).

The annual temperature was projected to increase by 2.2° C and 3.6° C on average in the watersheds under the RCP 4.5 and RCP 8.5 scenarios, respectively, compared to the historical average temperature of 12° C (Table 2), which was also consistent with the Kunkel et al. (2013). The overall predictions regarding the monthly maximum and minimum temperatures showed an increase, although a decrease was predicted for some watersheds. The variations of the minimum temperature across the study watersheds (ranges of the values or heights of the boxes in the box and whisker plots in Figure 3) were larger than those of the maximum temperature. The amount of the variations of the maximum temperature across the watersheds was relatively consistent over months, but the minimum temperature largely varied according to the watersheds during winter, indicating that the climate variability of the Ohio River watersheds would be more evident regarding the minimum temperature.

The monthly ensemble precipitation and temperature projection made by using the 35 GCMs for the entire 61 watersheds, as well as the "03232500" watershed that was selected as an example because of its representability in terms of location (the middle of the study watershed group) and size ($366 \text{ km}^2$: Figure 1(c)), are plotted in Figures 4 and 5, respectively. The amount of the variations of the projected precipitation did not change over time, but under RCP 8.5, the amount was larger than that under RCP 4.5 (Figures 4 and 5). Under the RCP 4.5 scenario, the GCMs predicted that the overall precipitation and temperature of the Ohio Basin watersheds would increase at the rates of 0.51 mm/decade and 0.28° C/decade, respectively, which correspond to the slopes of the linear trend lines of Figure 4, and that the rates increased to 1.25 mm/decade and 0.64° C/decade for precipitation and temperature, respectively, under the RCP 8.5 scenario. As seen in Figure 5, the variations of the precipitation ensemble were greater than those of the temperature ensemble, indicating the projection of precipitation is more susceptible to the selection of GCMs than is temperature projection.

### 3.2 Projected hydrologic changes

Monthly hydrographs of the hydrologic components that were generated using multiple GCMs and the behavioral parameters of the ABCD model were averaged to construct multi-parameter and multi-GCM ensemble streamflow hydrographs for each watershed (Figure 6; Tables 2 and 3). It is worth clarifying that the ranges and heights of the boxes





presented in the box and whisker plots that subsequently appear in this paper represent the projection variations across the 61 study watersheds.

The projections regarding the overall annual averages of PP (precipitation), QQ (total runoff or streamflow), DR (direct runoff), GW (groundwater), and ET (evapotranspiration) showed increases compared with those of the baseline (or historical) period under the RCP 4.5 and RCP 8.5 scenarios (Table 2). The projected increase rates of the hydrologic components including QQ, DR, and GW were greater than those for PP (6.8% and 8.8% for the RCP 4.5 and RCP 8.5 scenarios, respectively), which is in agreement with Fu et al. (2007), indicating that the precipitation changes were amplified in the runoff hydrographs (Table 2). The PET (potential ET) projections showed increases at rates similar to those of the TAV (average temperature) and PP, while the ET did not change as much as the TAV, PP, and PET because of the projected decreases of the infiltration and SS (soil water contents) (Table 2). The amount of available water in the watersheds was expected to increase by 1.6% and 1.5% for the RCP 4.5 and RCP 8.5 scenarios, respectively, implying that the overall amount of the available water in the Ohio River watersheds may not decrease in the future due to the projected increases in PP.

Simulated ensemble hydrographs showed unique watershed variations depending on the hydrologic component (Figure 6). For the cases of QQ, DR, GW, PET, ET, and WW (available water) (Figure 6), the spatial (across watersheds: heights of boxes, and the ranges between the maximum and minimum depths) and seasonal variations provided by the RCP 8.5 scenario and the far future (2070 to 2079) projection were generally somewhat larger than those provided by the RCP 4.5 scenario and the near future projection, respectively, indicating a greater uncertainty regarding far future hydrologic projections that are under extreme emission scenarios. The projections of DR showed greater seasonal variations than those of GW, which is in sound agreement with our watershed hydrology understandings, whereby the response of surface runoff to precipitation is more direct than that of groundwater to precipitation. The high seasonal variations found in the PET projections, ranging from 5 mm to 350 mm, were damped in the ET projections due to the interactions between soil particles and water that are expressed by the water-holding capacity of soil (Figure 6). The projections of SS and WW were widely and symmetrically distributed across the watersheds during each month, demonstrating the hydrologic variety of the selected watersheds. The annual and monthly watershed hydrology projections show that the multi-GCM and multi-parameter ensemble averages could provide reasonable descriptions of the overall hydrologic response of the Ohio River watersheds to climate projections.

The monthly projections of the multi-parameter and multi-model (or multi-GCM) ensembles regarding the hydrologic components were compared with the historical data to attain an understanding of the overall projected seasonal changes of the hydrology of the Ohio River watersheds (Table 3; Figures 7 and 8). QQ, DR, GW, and ET were projected to increase in all of the months, but SS was projected to decrease in most of the months, with the exception of January and February (Table 3 and Figure 7). WW of the watersheds was projected to decrease in June and July under the RCP 4.5 scenario, and in June, July, October, and November under the RCP 8.5 scenario. The increased rates of QQ and DR were larger than those of PP for all of the months, indicating that the amplified climate change impact on QQ is mainly attributed to the increases of DR





(Figures 7 and 8). In contrast with DR, the increased rates of GW were relatively low during winter, but they were higher than those of DR in summer, June, and July. The projected increase of the ET was relatively large in winter and spring, which corresponds to the temperature projection. The increased ET caused a decrease of SS, with the exception of January and February for which a small amount of ET was shown, implying an agricultural drought would be deepened in the watersheds; furthermore, water management needs to be more emphasized for maintaining the agricultural productivity of the Corn Belt areas in the future.

### 3.3 Uncertainty of hydrologic model parameter selection

The behavioral parameter values of the ABCD models developed for the 61 study watersheds were aggregated by the parameters to develop the overall parameter posterior distributions (Figure 9). The mode of the posterior distribution of parameter $a$ that is related to the infiltration capacity was the highest in the narrowest value range, and that of parameter $d$ for the control of groundwater flow showed the lowest mode with the widest value range. Considering the hydrologic meanings of $a$ and $d$ in the ABCD model (Thomas, 1981), such findings indicate infiltration excess mechanism is dominant in the watersheds, and the proportions of groundwater to streamflow are relatively variable and uncertain across the study watersheds. The posterior distribution of $b$ had a symmetric bell shape with a mode in the range from 200 to 500, meaning that the maximum monthly storage capacity in the watersheds is 350 mm on average. The parameter $c$ values were distributed around 0.1, ranging from 0.0 to 0.6, indicating that the groundwater contribution to streamflow is approximately 10%, but that it is also highly variable across the watersheds. The posterior distribution of $e$, introduced to adjust the ET values, was relatively symmetric around 1.0, but a tail was present from 1.5 to 2.0. For this study, the PET was estimated using the Hargreaves equation, followed by a calibration of the streamflow measurements using the parameter $e$. The posterior distribution showed that the Hargreaves equation that was used to calculate the PET provided sound estimations with respect to the water balance modeling for which the ABCD model was used; furthermore, though, the PET could also be overestimated or underestimated by as much as 50%, depending on the watershed.

Uncertainty in the projections of the multi-GCM and multi-parameter ensembles of hydrologic components was first quantified in the unit of depth by the study watersheds, then, it was normalized by dividing the depths by the precipitation depths for the purpose of a fair across-watershed comparison and quick approximations of uncertainty with known precipitation depths (Figure 10 and Figure 11). It is worth clarifying that the average values represent the overall uncertainty in the Ohio River watersheds, and the height of each box represents the across-watershed variations of the uncertainty in the box and whsker plots (Figure 10).

The overall average uncertainty in the monthly multi-parameter ensemble streamflow (QQ) projections for the 61 study watersheds varied from 9.2% (8.63 mm) to 13.4% (11.93 mm) of monthly precipitation depths under RCP 4.5 (Figure 10). Variations of the streamflow projection uncertainty amounts across the watersheds were relatively large in winter; no significant difference was found in the amounts of uncertainty between the QQ projections under RCP 4.5 and RCP 8.5





scenarios. The amount of uncertainty in the QQ projections was smaller than those in the DR projections, but they were larger than those in the GW projections, indicating that DR is more sensitive to parameter uncertainty than GW.

The PET projection showed a larger uncertainty compared to those of QQ, DR, or GW particularly in summer. The PET projections also showed great spatial variations across the latitudes between 36° 07'N and 42° 26'N within the Ohio River basin; alternatively, the uncertainty in the actual ET projections was relatively constant over all of the months and was somewhat larger in winter than in summer. ET was restricted by SS that was low in summer when soil was dry, and this regulated the variations of the uncertainty in the ET across the watersheds; moreover, compared with summer, the variation of ET was somewhat larger during winter when SS was relatively high. Uncertainty in SS did not largely vary by the seasons due to the water-holding capacity of the soil layers. Since WW mainly consisted of SS, the amounts and seasonal trends of their uncertainty are similar to each other.

### 3.4 Uncertainty in climate model selection

The selection of climate model was an order of magnitude more influential on uncertainty in the QQ, DR, and ET projections than that of parameter selection, but it was not always the case for GW and PET (Figures 10 to 12). In the case of QQ, the overall average uncertainty in the monthly multi-GCM ensemble projection ranged from 113% (99.3 mm) to 164% (160.4 mm) of monthly precipitation under RCP 4.5 (Figure 11). Uncertainty in the QQ projections was greater in winter than other seasons, and it was dominated by the GCM selection uncertainty (or the uncertainty in the ensemble projections) in DR. GW was relatively less responsive to GCM selection compared to QQ, DR, and ET (Figures 11 and 12). The influence of GCM selection on ET was far greater than that on the PET since ET is controlled by not only temperature but also the SS that is sensitive to GCM selection. The monthly variation patterns of uncertainty in PET were opposite to those of uncertainty in DR for both the multi-GCM and multi-parameter ensembles, implying that the influence of the PET uncertainty on DR projections is limited (Figures 10 and 11). Uncertainty in the WW projections due to GCM selection was 2 to 4 times larger than uncertainty in the PP projections, indicating the significance of GCM selection in a climate change impact analysis.

As the threshold values for the identification of behavioral parameter sets increased from 90.0% (a relatively conservative threshold for equifinality quantification) to 97.5% (a relatively liberal threshold), uncertainty in the GCM ensembles and its relative size to the uncertainty in the parameter ensembles increased exponentially (Figure 12). In the case of SS, parameter selection was more critical than GCM selection in all of the threshold cases. When relatively loose thresholds (i.e., 90.0% and 92.5%) were used, the selection of the hydrologic model parameters became more significant than GCM selection for the GW and PET projections, implying that the selection of hydrologic model parameters needs to be more careful than that for GCMs when soil moisture and groundwater are the concerns of a climate change impact study using the water balance model. Since ET is directly determined based on precipitation and direct runoff in the ABCD model, the corresponding uncertainty would become as significant as the uncertainty in the QQ projections (Figure 12). Overall, and depending on the

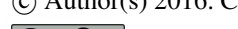



thresholds, GCM selection was 1.5 to 2 times more influential than parameter selection with respect to an assessment of the climate change impacts on the overall amount of available water in the watersheds.

The contribution of each GCM to the uncertainty in the GCM ensembles varied depending on the types of hydrologic components (Figure 13). Overall, the amounts of uncertainty contained in the climate change projections made by BCC, GCESS, CCCMA, CSIRO-QCCCE, and LASG-CESS for the Ohio River Basin watersheds were larger than those in the projections provided by INM, IPSL, and MIROC. Furthermore, the amount of uncertainty in precipitation projections of some GCMs was large while uncertainty in their temperature projections was small, and vice versa. The uncertainty amounts in the following three GCMs were relatively small for both of the climate variables in the Ohio River watersheds: CMCC-CMS, IPSL-CM5A-LR_1, and IPSL-CM5A-LR_4. The amounts of uncertainty in the GCM ensemble projections for the hydrologic components were highly correlated with those of the GCM-ensemble projections for precipitation rather than those for temperature (Figure 14). For example, the amount of uncertainty in the precipitation ensemble was related to those of QQ, DR, GW, and WW with correlation coefficients greater than 0.75, implying that the uncertainty in the precipitation ensemble was transferred to the hydrologic simulation. This finding also suggested that a greater effort needs to be invested in improving the projection accuracy of precipitation than temperature in a hydrologic analysis of climate change (Figures 14). PET was moderately correlated with TMX, TMN, and TAV, and this reflects the characteristics of the Hargreaves equation (Equation (5)) that was used for the PET calculation of this study.

# 4 Conclusions

This study demonstrated that the significance of GCM and hydrological parameter selection varied depending on the hydrologic components of interest and the thresholds used to identify the behavioral parameter sets in a hydrologic analysis of climate change. Streamflow and direct runoff projections were considerably affected by the uncertainty in multi-GCM ensembles, but soil moisture and groundwater projections were more responsive to the uncertainty in multi-parameter ensembles, implying that the selection of both GCMs and parameters should be carefully made to improve the reliability of a climate change impact analysis. The precipitation projection uncertainty was much more closely correlated to the uncertainty of the hydrological projections especially for runoff than that of the temperature projection, suggesting that the reliability of the precipitation projections made by GCMs need to be investigated for a robust hydrologic analysis of climate change. A newly proposed analysis strategy enabled to investigate the contributions of each GCM to uncertainty in a multi-GCM ensemble. Some of the GCMs produced more uncertainty in the hydrologic projections than others, but a corresponding investigation was beyond the scope of this study.

This study was implemented for 61 watersheds in the Ohio River basin to show the variability of the quantified amounts of uncertainty over different watersheds regarding size and location. Although a close relation might also exist between the unique landscape characteristics of the watersheds and the uncertainty amounts in the ensemble predictions, the matter has





been left to a future study. A total of 22 GCMs and their variants were considered in this study so that wide ranges of mathematical representations and the simulation strategies of climate change could be considered so that the largest uncertainty in the multi-GCM ensembles could be explored. Uncertainty associated with GCM selection was considerably large and greater than the amount of precipitation, indicating GCM selection is likely to substantially affect a hydrologic

analysis of climate change. Such a finding suggested that a map showing the ranges (uncertainty) and trends of the precipitation and temperature projections should be built using multiple GCMs, or hopefully all of them—which are used in the global scale climate projections for watersheds (e.g., 8- or 12-digit Hydrologic Unit Code watersheds)—to guide the field of hydrologic modeling for more effective GCM selection in the climate change studies regarding local watersheds.

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



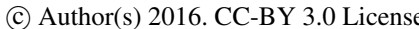










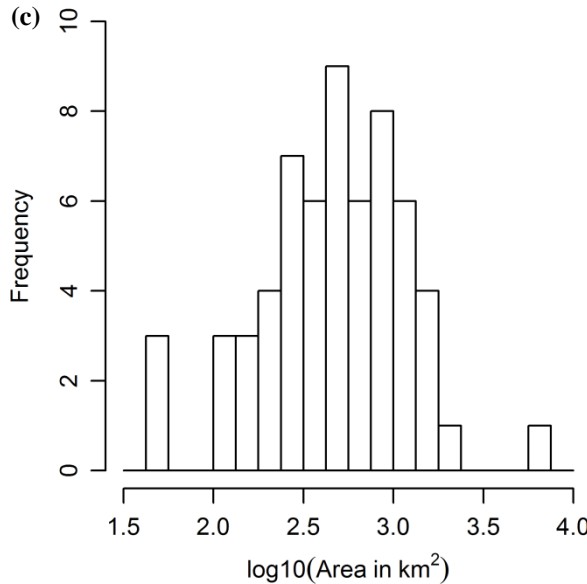

**Figure 1: Study watersheds in the Ohio River basin: (a) weather stations into which GCM data were downscaled and their Thiessen polygons, (b) stream networks and USGS gage stations where the ABCD model parameter was calibrated, and (c) variations in the sizes of the study watersheds selected for this study.**



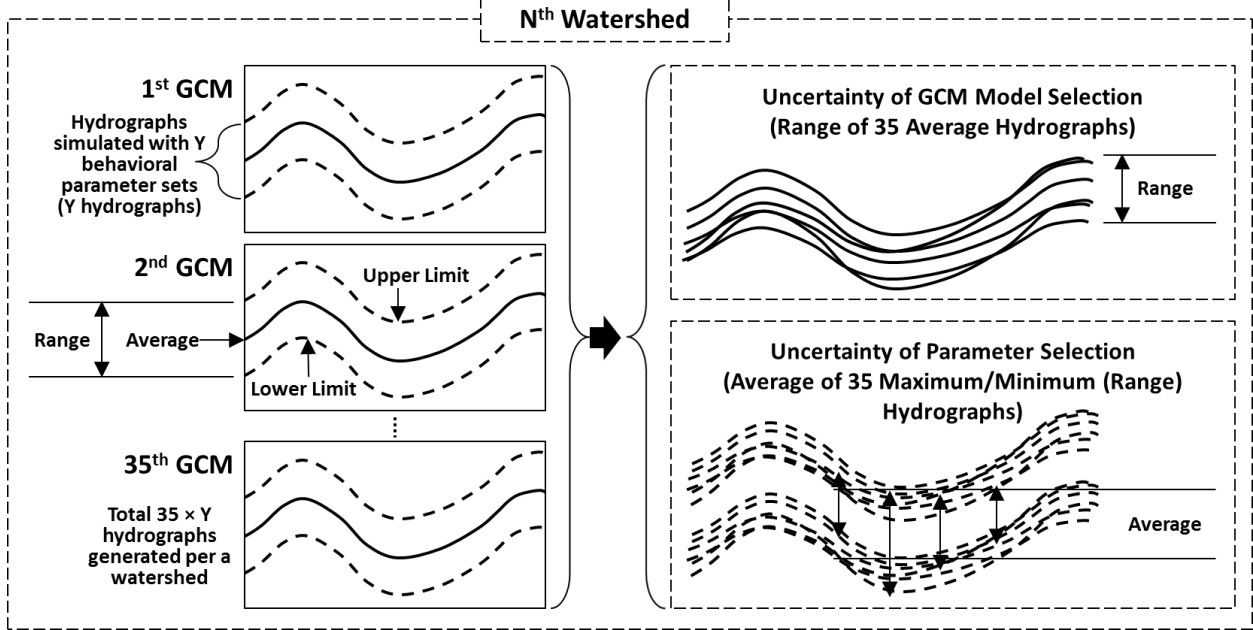

**Figure 2. Processes for the quantification of the uncertainty amounts in multi-GCM and multi-parameter ensembles.** *N* varies from 1 to 61, i.e., the number of the Ohio River watersheds selected for this study; *Y* represents the number of behavioral parameter sets identified for each watershed and therefore varies by the watershed.





**Figure 3: Overall monthly variations of the projected changes of precipitation (projected/historical) and temperature (projected/historical) across all of the study watersheds. a) and b): precipitation; c) and d): maximum temperature; e) and f): minimum temperature; a), c), and d): RCP 4.5; b), d), and f): RCP 8.5.**





**Figure 4: Multi-GCM, multi-parameter, and multi-watershed ensemble projections of the overall average precipitation and temperature of the Ohio River watersheds selected for this study from 2020 to 2099 (960 months). (a) and (c): RCP 4.5; and (b) and (d): RCP 8.5.**











**Figure 5: Monthly variations of the precipitation and temperature projected by the 35 GCMs for the "03232500" watershed. (a) and (e) near-future from 2030 to 2039 (120 months) under RCP 4.5; (b) and (f) far-future from 2070 to 2079 (120 months) under RCP 4.5; (c) and (g) near-future from 2030 to 2039 under RCP 8.5; and (d) and (h) far-future from 2070 to 2079 under RCP 8.5.**















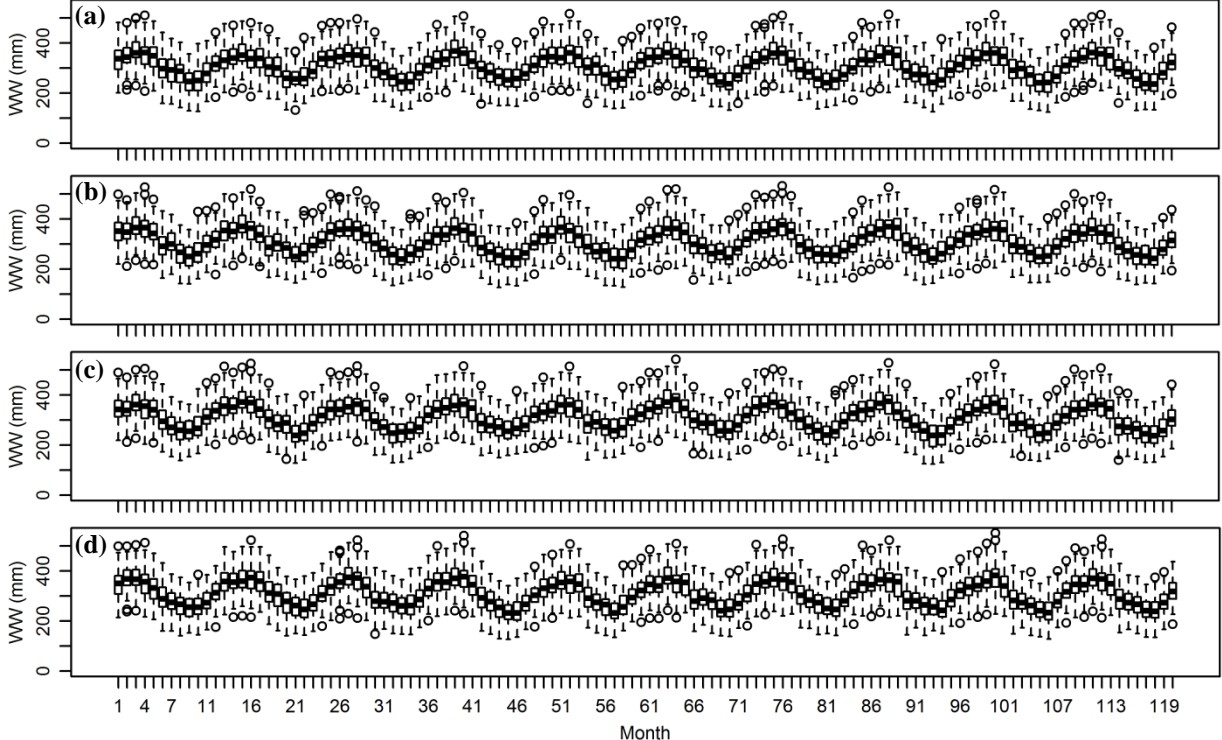

**Figure 6: Multi-parameter and multi-model ensemble projections for the hydrologic components (QQ, DR, GW, PET, ET, SS and WW) of the 61 Ohio Basin watersheds for the near-future (2030 to 2039, 120 months) and far-future (2070 to 2079, 120 months) periods under the RCP 4.5 and RCP 8.5 scenarios. For each hydrologic component, (a) and (b): near-future; (c) and (d): far-future; (a) and (c): RCP 4.5; and (b) and (d): RCP 8.5.**





**Figure 7: Multi-GCM, multi-parameter, and multi-watershed projections of the overall changes of the hydrologic components of the study watersheds: (a) RCP 4.5 and (b) RCP 8.5.**







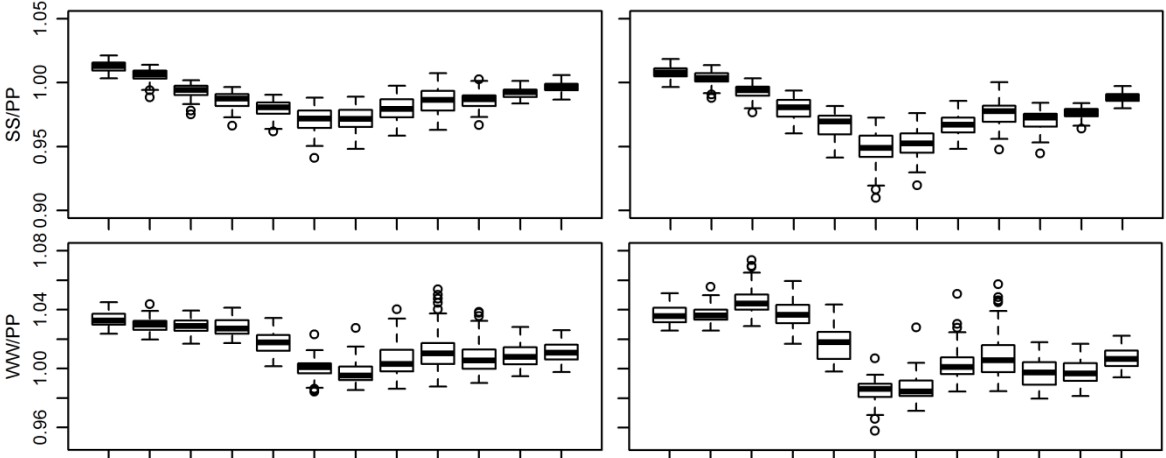

**Figure 8: Multi-GCM and multi-parameter ensemble predictions of the changes of the hydrologic components of the watersheds.**



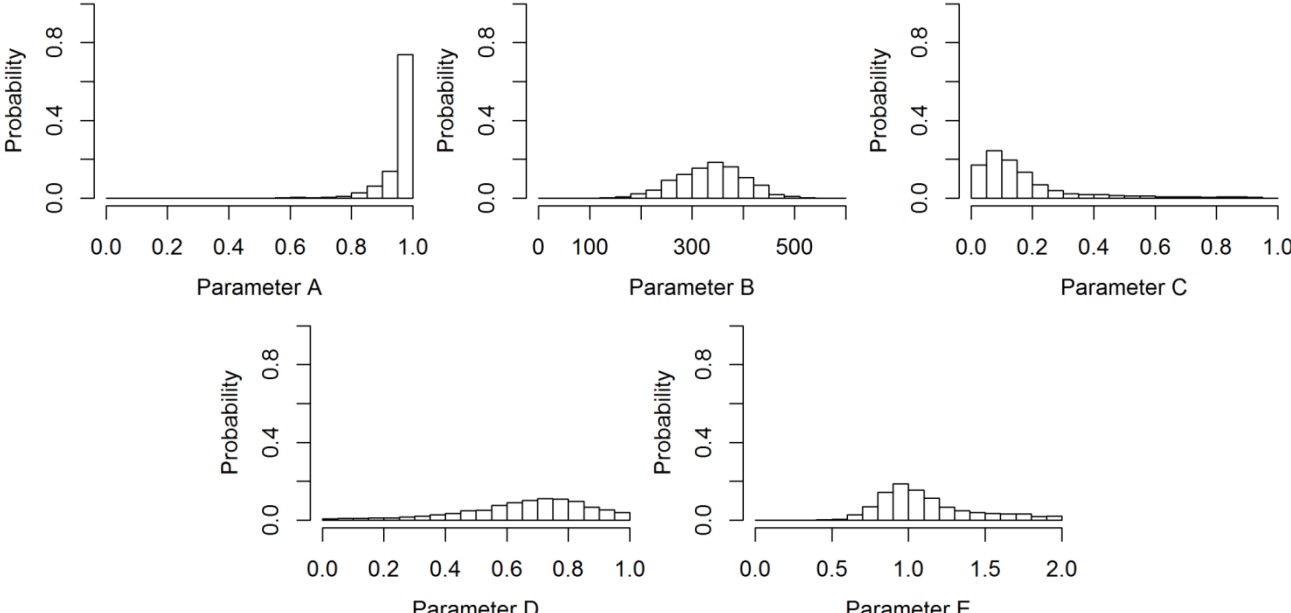

**Figure 9: Posterior distributions of the ABCD model parameters.**





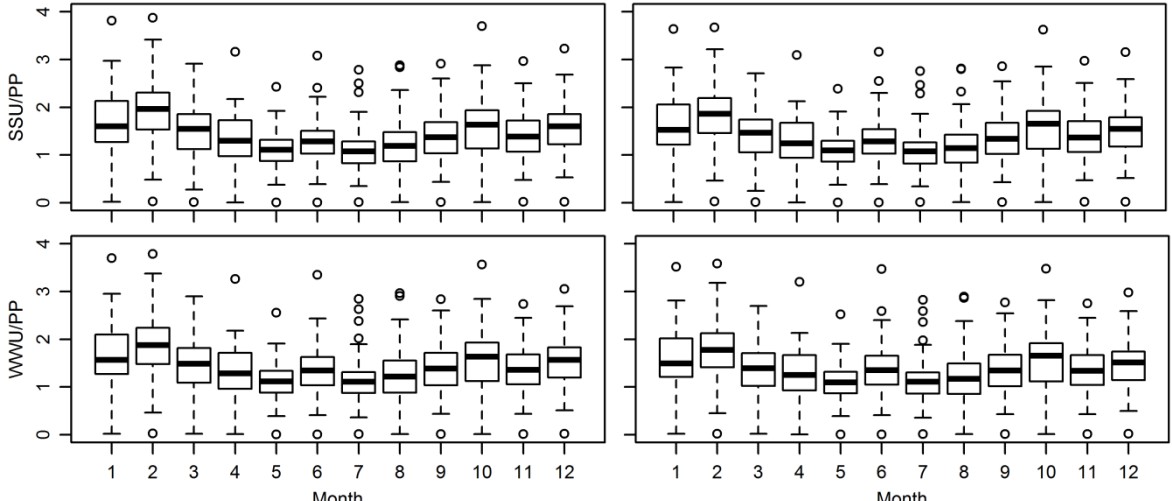

**Figure 10: Watershed-wide variations of the uncertainty amounts in the projected hydrological components due to parameter selection (uncertainty in multi-parameter ensemble projections). Results for the RCP 4.5 and RCP 8.5 scenarios are placed in the left and right columns, respectively. XXU signifies the uncertainty (U) of XX in the unit of XX (mm).**






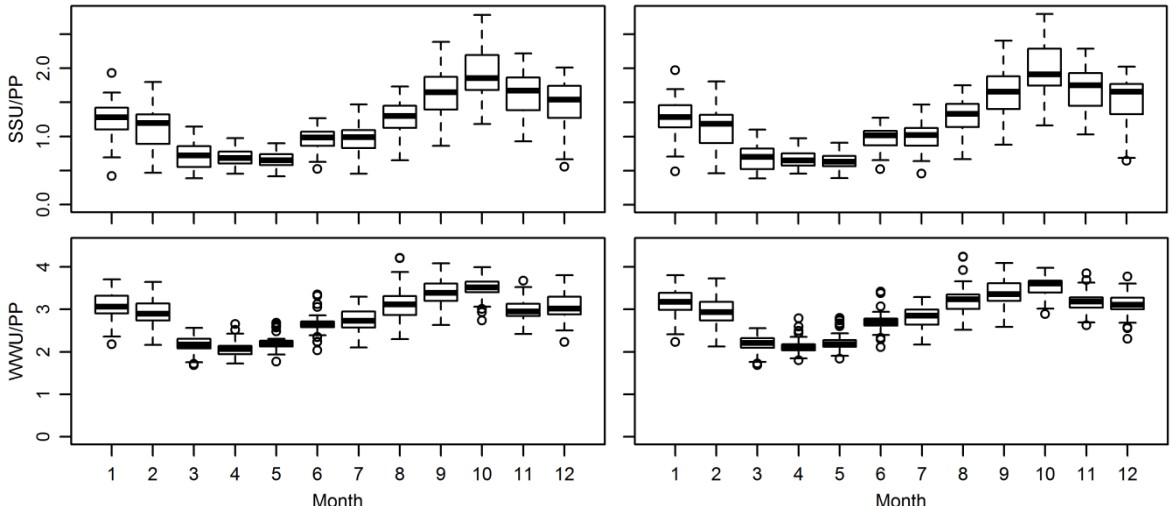

**Figure 11: Watershed-wise variations of the uncertainty amounts in the projected hydrological components due to GCM selection (uncertainty in multi-GCM ensemble projections). Results for the RCP 4.5 and RCP 8.5 scenarios are placed in the left and right columns, respectively. XXU signifies the uncertainty (U) of XX in the unit of XX (mm).**



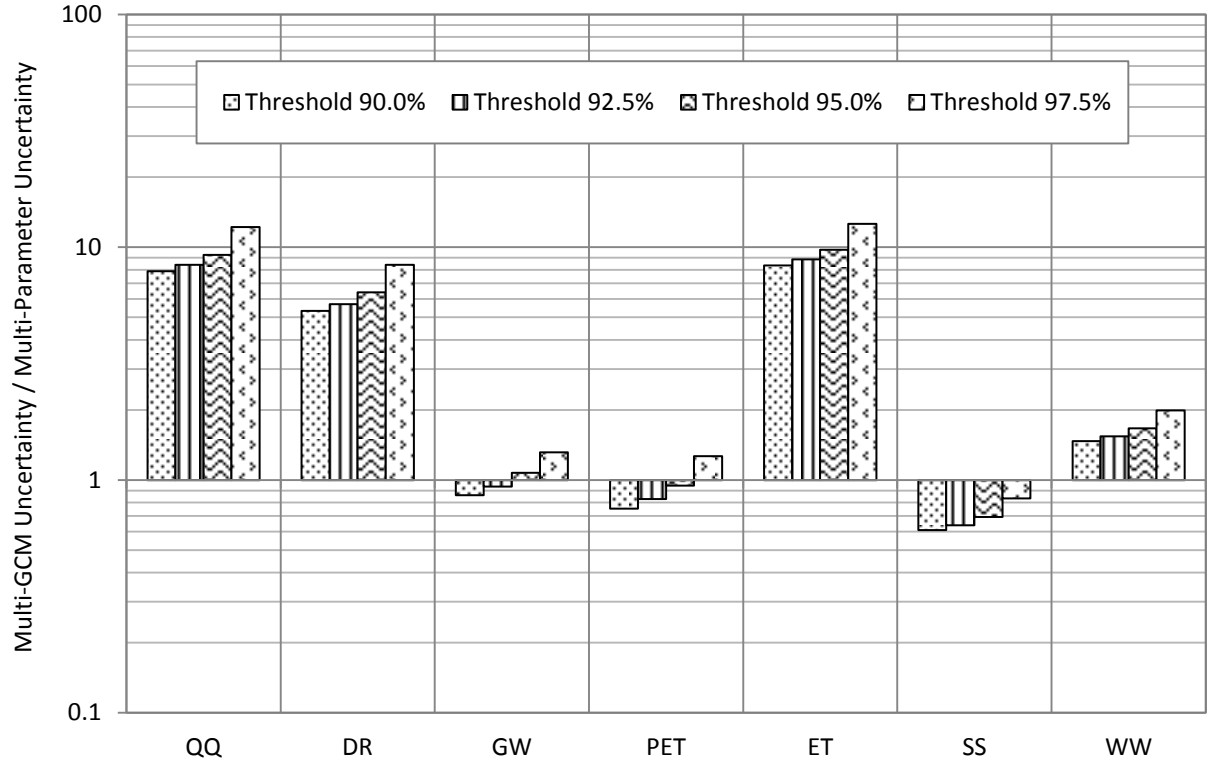

**Figure 12: Sensitivity of the quantified uncertainty in multi-parameter ensembles for hydrologic components.**





**Figure 13: Rankings of the GCM contributions to the uncertainty amounts in the multi-GCM-ensemble projections for climate variables and hydrologic components. Results for the RCP 4.5 and RCP 8.5 scenarios are placed in the left and right columns, respectively.**



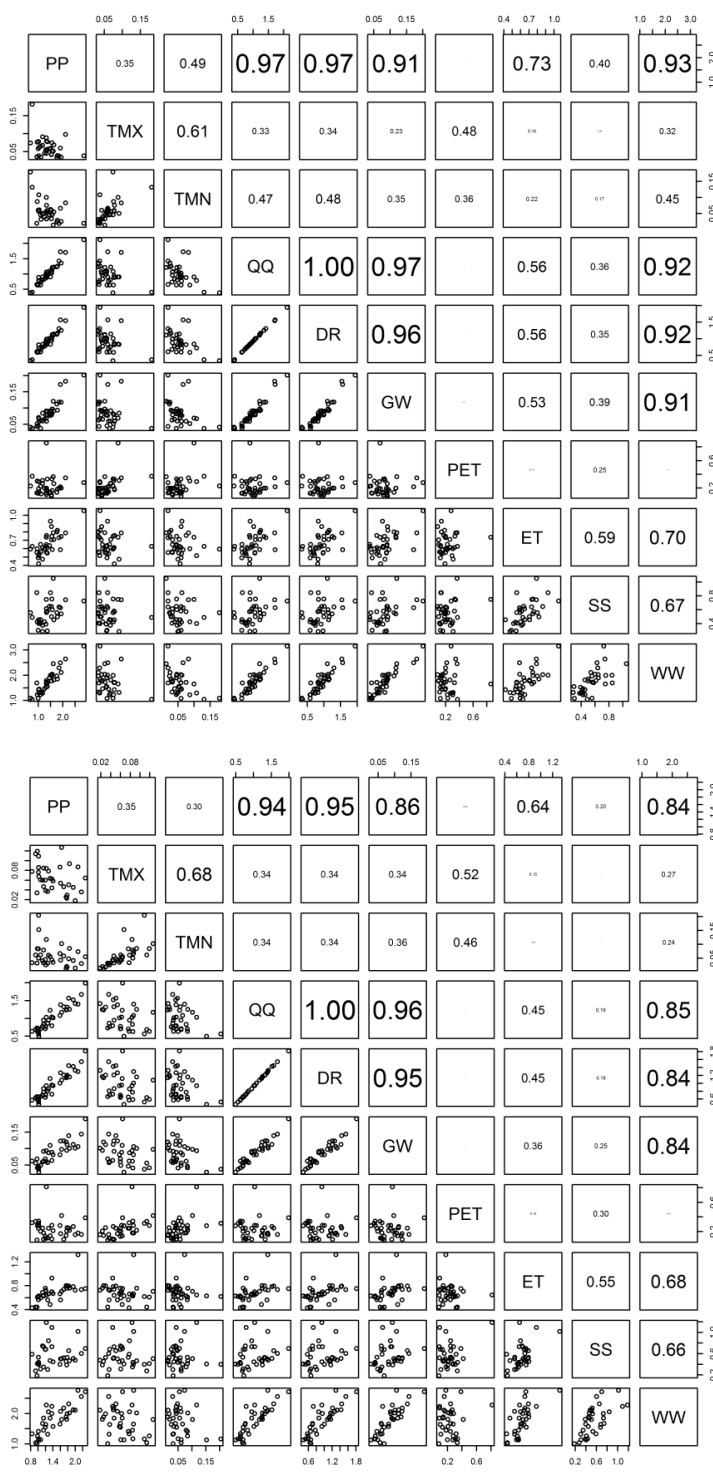

**Figure 14: Correlation between the uncertainty amounts in the projections for climate variables and the projections for hydrologic components: (a) RCP 4.5 and (b) RCP 8.5.**





**Table 1: The Coupled Model Intercomparison Project Phase 5 (CMIP5) GCM models, and their variants, used in this study (http://cmip-pcmdi.llnl.gov/cmip5/availability.html).**

| Model name | Realization number* | ID number | Institute ID | Resolution | Country |
|---|---|---|---|---|---|
| BCC-CSM1.1[1] | 1 | 1 | BCC | 64×128 | China |
| BCC-CSM1.1-m[1] | 1 | 2 | | 160×320 | |
| BNU-ESM[1] | 1 | 3 | GCESS | 64×128 | China |
| CanESM2[1] | 1,2,3,4,5 | 4-8 | CCCMA | 64×128 | Canada |
| CMCC-CMS[2] | 1 | 9 | CMCC | 96×192 | Italy |
| CMCC-CM[2] | 1 | 10 | | 240×480 | Italy |
| CNRM-CM5[2] | 1 | 11 | CNRM-CERFACS | 128×256 | France |
| CSIRO-Mk3.6.0[1] | 1 | 12 | CSIRO-QCCCE | 96×192 | Australia |
| FGOALS-g2[1] | 1 | 13 | LASG-IAP | 108×128 | China |
| FGOALS-s2[1] | 1,2,3 | 14-16 | LASG-CESS | | |
| GFDL-ESM2G[1] | 1 | 17 | NOAA GFDL | 90×144 | USA |
| GFDL-ESM2M[1] | 1 | 18 | | | |
| INM-CM4[1] | 1 | 19 | INM | 120×180 | Russia |
| IPSL-CM5A-LR[1] | 1,2,3,4 | 20-23 | IPSL | 96×96 | France |
| IPSL-CM5A-MR[1] | 1 | 24 | | 143×144 | |
| IPSL-CM5B-LR[1] | 1 | 25 | | 96×96 | |
| MIROC5[1] | 1,2,3 | 26-28 | MIROC | 128×256 | Japan |
| MIROC-ESM[2] | 1 | 29 | | 64×128 | |
| MIROC-ESM-CHEM[2] | 1 | 30 | | | |
| MPI-ESM-LR[2] | 1,2,3 | 31-33 | MPI-M | 96×192 | Germany |
| MPI-ESM-MR[2] | 1 | 34 | | | |
| MRI-CGCM3[2] | 1 | 35 | MRI | 160×320 | Japan |

1) calendar: 365 days (without a leap day)
2) calendar: Standard (with a leap day)

5 * "realization" number is used to distinguish among members of an ensemble typically generated by initializing a set of runs with different, but equally realistic, initial conditions.





**Table 2: Overall changes of the climate variables and hydrologic components projected by the multi-GCM, multi-parameter, and multi-watershed ensembles.**

| Variables | Statistics | Historical | RCP 4.5 | RCP 8.5 |
|---|---|---|---|---|
| Temperature (TAV) | Average (°C) | 12.0 | 14.2 | 15.6 |
| | Projected Change | - | 2.2% | 3.6% |
| Precipitation (PP) | Average (mm) | 1,084.9 | 1,159.0 | 1,180.0 |
| | Projected Change | - | 6.8% | 8.8% |
| Total runoff (QQ) | Average (mm) | 402.5 | 442.7 | 451.9 |
| | Projected Change | - | 10.0% | 12.3% |
| QQ/PP | Projected Change | 37.1% | 38.2% | 38.3% |
| Direct runoff (DR) | Average (mm) | 337.8 | 372.2 | 380.2 |
| | Projected Change | - | 10.2% | 12.6% |
| DR/QQ | Projected Change | 83.9% | 84.1% | 84.1% |
| Groundwater (GW) | Average (mm) | 62.1 | 67.8 | 69.1 |
| | Projected Change | - | 9.2% | 11.3% |
| Evapotranspiration (ET) | Average (mm) | 682.4 | 716.5 | 728.0 |
| | Projected Change | - | 5.0% | 6.7% |
| ET/PP (= 1 − QQ/PP) | Projected Change | 62.9% | 61.8% | 61.7% |
| Potential ET (PET) | Average (mm) | 1,085.3 | 1,154.4 | 1,183.1 |
| | Projected Change | - | 6.4% | 9.0% |
| Soil Moisture (SS) | Average (mm) | 2,558.4 | 2,535.6 | 2,510.4 |
| | Projected Change | - | -0.9% | -1.9% |
| Available Water (WW) | Average (mm) | 3,634.8 | 3,691.2 | 3,687.6 |
| | Projected Change | - | 1.6% | 1.5% |





**Table 3: Projected percentage changes (projection period of 2020 to 2099 vs. historical period of 1980 to 2000) of the hydrologic components of the study watersheds: the multi-GCM, multi-parameter and multi-watershed ensemble projections for the Ohio River watersheds (unit: dimensionless ratio, projected data/historical data).**

| Item | RCP | Jan | Feb | Mar | Apr | May | Jun | Jul | Aug | Sep | Oct | Nov | Dec |
|---|---|---|---|---|---|---|---|---|---|---|---|---|---|
| PP | 4.5 | 8.49 | 9.86 | 9.64 | 11.55 | 7.39 | 3.72 | 3.64 | 5.83 | 6.84 | 5.25 | 5.64 | 6.49 |
|  | 8.5 | 12.69 | 15.2 | 17.04 | 14.81 | 8.50 | 1.58 | 3.98 | 8.23 | 8.00 | 4.25 | 5.46 | 9.63 |
| QQ | 4.5 | 10.67 | 10.83 | 10.45 | 12.62 | 10.09 | 6.15 | 7.67 | 12.39 | 13.01 | 10.54 | 7.33 | 7.69 |
|  | 8.5 | 13.69 | 14.73 | 18.28 | 17.47 | 11.20 | 2.46 | 6.36 | 16.24 | 13.61 | 7.93 | 7.00 | 7.94 |
| DR | 4.5 | 11.66 | 11.32 | 10.64 | 12.96 | 10.03 | 5.40 | 7.50 | 13.45 | 13.72 | 10.54 | 7.00 | 7.58 |
|  | 8.5 | 15.03 | 15.5 | 18.91 | 17.78 | 10.72 | 0.53 | 5.85 | 18.03 | 14.09 | 7.26 | 6.62 | 7.88 |
| GW | 4.5 | 1.43 | 5.71 | 8.12 | 10.14 | 10.19 | 8.87 | 8.46 | 9.88 | 10.69 | 10.87 | 9.24 | 8.60 |
|  | 8.5 | 1.79 | 7.67 | 13.06 | 15.07 | 13.47 | 9.98 | 8.74 | 11.31 | 11.81 | 10.61 | 9.00 | 8.67 |
| PET | 4.5 | 12.9 | 12.65 | 10.83 | 8.28 | 5.96 | 5.12 | 4.11 | 4.22 | 5.65 | 7.62 | 10.16 | 10.41 |
|  | 8.5 | 18.39 | 16.43 | 13.71 | 11.18 | 8.70 | 7.70 | 6.19 | 5.92 | 7.71 | 11.57 | 14.74 | 15.03 |
| ET | 4.5 | 5.54 | 8.07 | 8.32 | 10.45 | 5.62 | 2.68 | 2.44 | 4.27 | 5.43 | 3.86 | 5.01 | 5.64 |
|  | 8.5 | 11.34 | 16.2 | 14.87 | 12.09 | 6.76 | 1.18 | 3.35 | 6.33 | 6.75 | 3.28 | 4.91 | 10.9 |
| SS | 4.5 | 1.28 | 0.51 | -0.66 | -1.38 | -2.06 | -2.93 | -2.81 | -2.08 | -1.36 | -1.42 | -0.78 | -0.37 |
|  | 8.5 | 0.78 | 0.30 | -0.72 | -2.03 | -3.30 | -5.23 | -4.80 | -3.28 | -2.34 | -2.98 | -2.41 | -1.16 |
| WW | 4.5 | 3.36 | 2.98 | 2.88 | 2.81 | 1.69 | -0.01 | -0.18 | 0.61 | 1.21 | 0.75 | 0.88 | 1.13 |
|  | 8.5 | 3.67 | 3.68 | 4.62 | 3.68 | 1.67 | -1.57 | -1.34 | 0.38 | 0.86 | -0.25 | -0.23 | 0.75 |

