# Peer review of "Comparison of uncertainty in multi-parameter and multi-model ensemble hydrologic analysis of climate change"

_Hydrology and Earth System Sciences, 2016_

## Referee Comment (RC1) · Anonymous Referee #1 · 20 Oct 2016

p4 L16 please explain more why choosing the ABCD model and not a more process based model, as your investigation is aiming towards hydrological processes a process based simple hydrological model should be chosen

p4 L23 Typo should be removed

p12 L31 enhance the section of possible uncertainties if hydrological models and ensemble prediction. for me it becomes not clear what has essential influences in your research and how you estimate that.

---

## Referee Comment (RC2) · Anonymous Referee #2 · 20 Oct 2016

In this manuscript, the authors quantify the uncertainty in multi-parameter and multi-model ensemble hydrologic analysis of climate change using 61 Ohio River watersheds. The authors show from their results that the relative contribution of uncertainty in multi-GCM ensembles can be an order of magnitude larger than that of multi-parameter ensembles when predicting direct run-off. Evaluating groundwater and soil moisture, multi-parameter ensembles show to be the largest driver of uncertainties. The authors demonstrate within their study a "novel" framework for uncertainty-analysis which could be applied in other catchments.

All in all, the paper addresses relevant scientific questions within the scope of HESS. Although I do not think the approach for evaluating uncertainties they suggest is very

novel, some substantial conclusions can be drawn from the research, which might be of interest for the water resources research community. I would therefore support the manuscript for publication but with substantial revisions taking into account the following general and technical comments/suggestions:

General comments: 1. In this study the authors used ABCD, a mathematical model rather than a process-based model. I wonder whether using a process-based hydrological model wouldn't solve already large part of the equifinality problem mentioned by the authors. Looking into the physical boundary values for each of the sub-parameters, the physical processes themselves, and calibrating/validating the sub-hydrological results should already for a large part solve this equifinality issue, resulting in only one or perhaps a few sets of parameters that describe the overall hydrological system best.

2. The authors mention that in the ABCD model soil water content is proportional to the evapotranspiration opportunity and that this exponentially increases with the potential evapotranspiration rate: from a physical point of view, shouldn't this be the other way around? Potential evapotranspiration being driven/(or limited) by the availability of soil moisture?

3. Does a simpler model (page 3 – line 10) have less uncertainty because they have less parameters incorporated? Or is the uncertainty less visible? Or: Does a more complex model lead per definition to higher uncertainty?

4. I doubt whether you can consider the different GCMs to be independent, since quite a few of them a highly related to each other. Could you elaborate how to deal with this?

5. The actual discussion part is relatively small. I would suggest the authors to elaborate a bit more on the impacts of their findings. But also to discuss any uncertainties/limitations of their conducted research.

6. From the text it did not became clear to me how the statistical downscaling of the GCM climate projections was done. Please elaborate a bit more on that.

[Figure]

7. Overall, the study is quite wordy and different definitions are being used throughout the text. Try to be consistent in the use of definitions and remove redundant text.

8. Too many figures are included in the manuscript and most of the figures are too difficult to grasp. Try to simplify and diminish the number of figures shown.

Technical comments:

- P3-L3: "Many different models": Out of many the authors name only three models, why not mention the others widely used: e.g. WaterGAP, PCR-GLOBWB, H08, LPJmL, etc?

- P4-L2: "22... 35 GCMs": How were 35 GCMs derived from the initial 22 ones? What was the selection procedure applied here?

- P4-L16: "locations": What do you mean here?

- P5-L27: what is meant with the long-term monthly hydrologic response?

- P5-L28: WW is an often used indicator for water withdrawals. Better use WA here.

- P7-L11: "maximum and minimum values": please specify where these values refer to

- P7-L23 (formula): where do the subscripts mentioned in the formula come from? Cannot find their meaning in the text.

- P8-L19: "03232500": Doesn't this watershed have a name?

- P9-L5-6: I do not understand how to interpret these values: are these the average projected increase rates across all hydrologic components?

- P9-L7-8: "indicating...runoff hydrographs": What could be a reason for this observation? In most models you see that precipitation changes are buffered towards runoff estimates.

- P9-L11-13: "Implying...PP": Where does this water go to then? Increased ET? Higher soil moisture? Or less low-period flows?

- P9-L28: "multi-model(multi-GCM): just give it one name: multi-GCM.

- P10-L9: I do not fully understand the idea behind the overall parameter posterior distribution. What is the added value to aggregate the results over all the watersheds?

- P10-L24: "Unit of depth": Not clear what is meant with this

- P11-L5: "relatively constant .. than in summer": sounds like a contradiction

- P11-L26: Not clear to me whether a threshold of 97.5% is relatively loose or conservative.

- P11-L29: In the ABCD model soil moisture and groundwater are 'rest-products', isn't it straightforward then that the impact of model-parameterization is larger than the impact of uncertainty in GCM-input?

- P11-L30: How can ET be directly be determined by direct runoff? Shouldn't this be the other way around: uncertainty in QQ being driven by uncertainty in ET?

- P12-L10: So would you say that, in the Ohio River basin, precipitation is a larger determinant for water availability than Temperature?

- P12-L13-14: "This finding…climate change": This is an observation that hold for this basin specifically. Does it also hold for other types of catchments, e.g. the temperature dominated ones?

- P12-L19-20: "and the thresholds…climate change": Incomplete sentence

- P13-L1-3: "A total of… explored": Fuzzy sentence, please rephrase.

- P13-L3-4: "Uncertainty associated…amount of precipitation": I don't understand this statement. Please clarify.

- P13-L7: How about the regional scale climate projections?

- Fig1: Is it necessary to present the area size in terms of log10?

- Fig5: These sub-figures are very difficult to compare. Would it be possible to make 2 figures (one for precip and one for temp) showing the ltm-hydrographs (12 months) for the different scenarios: * current conditions * near future under rcp4.5 and rcp8.5 * far future under rcp4.5 and rcp8.5

- Fig 6: see comment figure 5.

- Fig 7: Does it really make sense to show a multi-GCM, multi-parameter, multi-watershed projection?

- Fig 9: Are these values for all watersheds?

- Fig 12: Could you explain why for GW and PET the sign changes when the threshold increases? In lower thresholds multi-parameter uncertainty shows to be more important, whilst in high thresholds multi-GCM is a more important determinant for the uncertainty in outcomes.

- Fig 13: Legend of this figure is difficult to interpret. Perhaps give a figurative example to clarify.

---

## Author Comment (AC1) · 16 Nov 2016

Comments R1.1: p4 L16 please explain more why choosing the ABCD model and not a more process based model, as your investigation is aiming towards hydrological processes a process based simple hydrological model should be chosen Response to Comment R1.1: The objective of this study is to compare the impacts of climate model and hydrologic model parameter selections on the projections of hydrologic components. Considering the spatiotemporal extent (61 watersheds over the next 80 years) of hydrologic projections to be made for the study, we looked for a simulation model that is parsimonious while being capable of representing hydrologic components of interest, including direct runoff, soil water, evapotranspiration, and groundwater. The

[Figure]

ABCD model well satisfies our needs for this study, and the model has been success-ful in hydrological analyses such as Sankarasubramanian and Vogel (2002), Kirshen et al. (2005), and Martinez and Gupta (2010). For the purpose of clarification to this comment, we will add a description and reasoning on the model selection process in the section "2.3 Hydrologic model", which reads "The ABCD model was selected due to its parsimonious structure requiring only five parameters and allowing computation-ally affordable simulation of hydrologic components of interest including direct runoff, soil water, evapotranspiration, and groundwater". References Sankarasubramanian, A. and Vogel, R.M., 2002. Annual hydroclimatology of the United States. Water Re-sources Research, 38(6), WR000619.1-12. Kirshen, P., McCluskey, M., Vogel, R. and Strzepek, K., 2005. Global analysis of changes in water supply yields and costs under climate change: a case study in China. Climatic Change, 68(3), pp.303-330. Martinez, G.F. and Gupta, H.V., 2010. Toward improved identification of hydrological models: A diagnostic evaluation of the "abcd" monthly water balance model for the conterminous United States. Water Resources Research, 46(8), W08507.1-21.

Comments R1.2: p4 L23 Typo should be removed Response to Comment R1.2: The typo of "Speical" will be fixed to "Special".

Comments R1.3: p12 L31 enhance the section of possible uncertainties if hydrological models and ensemble prediction. for me it becomes not clear what has essential influences in your research and how you estimate that Response to Comment R1.3: As suggested, we will further describe how the proposed uncertainty analysis strategy helps the selection of climate models, which reads "The GCM uncertainty contributions quantified using the proposed analysis strategy would be a useful information and indicator to screen GCMs in creating improved ensemble hydrologic projections". In addition, the last sentence of the paragraph will be modified to "Some of the GCMs produced more uncertainty in the hydrologic projections than did others, but an investigation on the relationship between the characteristics of climate models and their contributions to the overall uncertainty was beyond of the scope of this

study." To further clarify the method used to estimate uncertainty, we will add the following sentence at the end of "2.5 Quantification of uncertainty in multi-parameter and multi-GCM ensemble": "The equation calculates the overall variation ranges (UˆQ (ãĂŰGCMãĂŮ_(âĹĂx âĹĹ S) )) of climate variable and hydrologic component projections made using all climate models (âĹĂx âĹĹ S) then subtracts the variation ranges (UˆQ (ãĂŰGCMãĂŮ_(x âĹĽ S) )) of the projections made excluding a specific climate model (x âĹĽ S) from the overall variation ranges to quantify the uncertainty contribution of the specific model." (See equations in an attached pdf file)

Please also note the supplement to this comment:
http://www.hydrol-earth-syst-sci-discuss.net/hess-2016-160/hess-2016-160-AC1-supplement.pdf
* * *

---

## Author Comment (AC2) · 16 Nov 2016

Comments R2.1: In this manuscript, the authors quantify the uncertainty in multi-parameter and multimodel ensemble hydrologic analysis of climate change using 61 Ohio River watersheds. The authors show from their results that the relative contribution of uncertainty in multi-GCM ensembles can be an order of magnitude larger than that of multiparameter ensembles when predicting direct run-off. Evaluating ground-water and soil moisture, multi-parameter ensembles show to be the largest driver of uncertainties. The authors demonstrate within their study a "novel" framework for uncertainty-analysis which could be applied in other catchments. All in all, the paper addresses relevant scientific questions within the scope of HESS. Although I do

Interactive
comment

not think the approach for evaluating uncertainties they suggest is very novel, some substantial conclusions can be drawn from the research, which might be of interest for the water resources research community. I would therefore support the manuscript for publication but with substantial revisions taking into account the following general and technical comments/suggestions: Response to Comment R2.1: We appreciate your positive and constructive comments. You have added a great value to our manuscript and improved the quality of work.

Comments R2.2: General comments: 1. In this study the authors used ABCD, a mathematical model rather than a process-based model. I wonder whether using a process-based hydrological model wouldn't solve already large part of the equifinality problem mentioned by the authors. Looking into the physical boundary values for each of the sub-parameters, the physical processes themselves, and calibrating/validating the sub-hydrological results should already for a large part solve this equifinality issue, resulting in only one or perhaps a few sets of parameters that describe the overall hydrological system best. Response to Comment R2.2: We agree with your point. A process-based model may better regulate parameters by directly relating physical features of a watershed to parameter values. However, a hydrologic model is still a simplification of the reality, thus a certain level of conceptualization or lumping would be necessary in representing hydrologic processes happening at a certain spatiotemporal scale. In addition, a process-based model tends to have a large number of parameters to describe many details of hydrologic processes, which may add uncertainty to the outputs. Her and Chaubey (2015) demonstrated that increasing the number of calibration parameters might not reduce the amount of modeling output uncertainty but would decrease uncertainty in the calibration parameters. As mentioned in the second paragraph of "4. Conclusion", the physical characteristics of a watershed might be closely associated with the amount of uncertainty, and we might be able to reduce the level of equifinality of the hydrologic modeling by directly relating values of parameters to measurements made at fields. However, we leave this for our next study as it is beyond the scope of this study, and it does not affect the conclusion of this study. References Her, Y. and

Chaubey, I., 2015. Impact of the numbers of observations and calibration parameters on equifinality, model performance, and output and parameter uncertainty. Hydrological Processes, 29(19), pp.4220-4237.

Comments R2.3: 2. The authors mention that in the ABCD model soil water content is proportional to the evapotranspiration opportunity and that this exponentially increases with the potential evapotranspiration rate: from a physical point of view, shouldn't this be the other way around? Potential evapotranspiration being driven/(or limited) by the availability of soil moisture? Response to Comment R2.2: The ABCD model was designed to simulate the responses of hydrologic components such as soil moisture, direct runoff, and groundwater to rainfall and potential evapotranspiration (PET) (Thomas, 1981; Sankarasubramanian and Vogel, 2002; Martinez and Gupta, 2010). PET is a function of weather/climate variables rather than physical features of a watershed; thus PET is not limited by the amount of soil moisture available. In the ABCD model, actual evapotranspiration is regarded as the difference between total runoff and rainfall, and total runoff is a function of soil moisture (SS). Thus, ET (or more precisely, actual ET) is controlled by SS in ABCD. References Thomas, H.: Improved methods for national water assessment, Report WR15249270, US Water Resource Council, Washington, DC, 1981. Sankarasubramanian, A. and Vogel, R.M., 2002. Annual hydroclimatology of the United States. Water Resources Research, 38(6), WR000619.1-12. Martinez, G.F. and Gupta, H.V., 2010. Toward improved identification of hydrological models: A diagnostic evaluation of the "abcd" monthly water balance model for the conterminous United States. Water Resources Research, 46(8), W08507.1-21.

Comments R2.4: 3. Does a simpler model (page 3 – line 10) have less uncertainty because they have less parameters incorporated? Or is the uncertainty less visible? Or: Does a more complex model lead per definition to higher uncertainty? Response to Comment R2.4: According to Her and Chaubey (2015), a simpler model that has a less number of calibration parameters does not necessarily have smaller amount of uncertainty in its output, even though it has larger uncertainty in its parameters (equifinality).

On the other hand, a more complicated model with a larger number of calibration parameters has greater amount of equifinality. As mentioned in the manuscript, therefore, a simpler model would be preferred as long as it can satisfy modeling needs since it requires less input data and calibration parameters. References Her, Y. and Chaubey, I., 2015. Impact of the numbers of observations and calibration parameters on equifinality, model performance, and output and parameter uncertainty. Hydrological Processes, 29(19), pp.4220-4237.

Comments R2.5: 4. I doubt whether you can consider the different GCMs to be independent, since quite a few of them a highly related to each other. Could you elaborate how to deal with this? Response to Comment R2.5: We are aware of that there are GCMs correlated to each other. However, we included all of them in the analysis without considering the correlation structure since the objective of this study is investigating the range of the variations in the projections rather than developing a single ensemble projection. The development of an ensemble projection may require considering the correlation structure by assigning different weights to climate models based on their similarity to each other or performance in reproducing historical climate variables, but quantifying the variation ranges of projected climate variables across GCMs does not.

Comments R2.6: 5. The actual discussion part is relatively small. I would suggest the authors to elaborate a bit more on the impacts of their findings. But also to discuss any uncertainties/limitations of their conducted research. Response to Comment R2.6: Following the suggestions of the reviewer, we will add discussions on the utility of the method proposed to quantify the contribution of each climate model to the overall uncertainty, which reads "The GCM uncertainty contributions quantified using the proposed analysis strategy would be a useful information and indicator to screen GCMs in improving ensemble hydrologic projections". In addition, we will mention that "On the whole, the results demonstrate how a subjective selection of climate models and hydrologic model parameter values can influence on hydrologic analysis regarding climate change and highlight the importance of quantitative analysis of uncertainty in

hydrologic analysis of climate change" in the conclusion. To clarify the limitations of this study, we will add that "In addition, this study included watersheds whose hydrologic responses to rainfall and temperature could be reproduced and explained with a simple water balance model, ABCD, thus the results might not be universally applicable to any watershed", and "We employed a statistical downscaling method developed by Ho et al. (2012) in this study, thus the use of different downscaling methods may lead to different analysis results (Chen et al., 2011)" in the conclusion section. References Chen, J., Brissette, F.P., Poulin, A. and Leconte, R., 2011. Overall uncertainty study of the hydrological impacts of climate change for a Canadian watershed. Water Resources Research, 47(12). Ho, C.K., Stephenson, D.B., Collins, M., Ferro, C.A. and Brown, S.J., 2012. Calibration strategies: a source of additional uncertainty in climate change projections. Bulletin of the American Meteorological Society, 93(1), p.21.

Comments R2.7: 6. From the text it did not became clear to me how the statistical downscaling of the GCM climate projections was done. Please elaborate a bit more on that. Response to Comment R2.7: In the revised manuscript, we will add a description on the statistical downscaling method and processes used in our study, which reads "The bias correction approach used in this study assumes that discrepancies between observed and modeled climate variables do not change over time or in the future. Thus, future observables can be directly predicted based on historical observations using a transfer function mapping simulated climate onto observations. The transfer function was estimated by matching the predicted future probability distributions of climate variables to their empirical (historical) distributions (Eq. 1)."

Comments R2.8: 7. Overall, the study is quite wordy and different definitions are being used throughout the text. Try to be consistent in the use of definitions and remove redundant text. Response to Comment R2.8: We critically reviewed the manuscript and identified redundant texts. In the revised manuscript, we will remove the redundancy and use consistent terms.

Comments R2.9: 8. Too many figures are included in the manuscript and most of the

figures are too difficult to grasp. Try to simplify and diminish the number of figures shown. Response to Comment R2.9: We identified less important figures, including four sub-figures of Figure 5 for RCP 4.5 and fourteen sub-figure for RCP 4.5 of Figure 6, and they will be removed from the manuscript.

Comments R2.10: P3-L3: "Many different models": Out of many the authors name only three models, why not mention the others widely used: e.g. WaterGAP, PCR-GLOBWB, H08, LPJmL, etc? Response to Comment R2.10: As suggested, we further reviewed the literature related to hydrologic analysis of climate change and added two more models (WaterGAP and LPJmL) to the manuscript as they have been widely used in the past 10 years. References Elliott, J., Deryng, D., Müller, C., Frieler, K., Konzmann, M., Gerten, D., Glotter, M., Flörke, M., Wada, Y., Best, N. and Eisner, S., 2014. Constraints and potentials of future irrigation water availability on agricultural production under climate change. Proceedings of the National Academy of Sciences, 111(9), pp.3239-3244. Langerwisch, F., Rost, S., Gerten, D., Poulter, B., Rammig, A. and Cramer, W., 2013. Potential effects of climate change on inundation patterns in the Amazon Basin. Hydrology and Earth System Sciences, 17(6), pp.2247-2262. Olesen, J.E., Carter, T.R., Diaz-Ambrona, C.H., Fronzek, S., Heidmann, T., Hickler, T., Holt, T., Minguez, M.I., Morales, P., Palutikof, J.P. and Quemada, M., 2007. Uncertainties in projected impacts of climate change on European agriculture and terrestrial ecosystems based on scenarios from regional climate models. Climatic Change, 81(1), pp.123-143. Schmied, H.M., Adam, L., Eisner, S., Fink, G., Flörke, M., Kim, H., Oki, T., Portmann, F.T., Reinecke, R., Riedel, C. and Song, Q., 2016. Variations of global and continental water balance components as impacted by climate forcing uncertainty and human water use. Hydrology and Earth System Sciences, 20(7), pp.2877-2896. Schewe, J., Heinke, J., Gerten, D., Haddeland, I., Arnell, N.W., Clark, D.B., Dankers, R., Eisner, S., Fekete, B.M., Colón-González, F.J. and Gosling, S.N., 2014. Multimodel assessment of water scarcity under climate change. Proceedings of the National Academy of Sciences, 111(9), pp.3245-3250. Döll, P. and Zhang, J., 2010. Impact of climate change on freshwater ecosystems: a global-scale analysis of ecologically relevant river flow

alterations. Hydrology and Earth System Sciences, 14(5), pp.783-799.

Comments R2.11: P4-L2: "22. . . 35 GCMs": How were 35 GCMs derived from the initial 22 ones? What was the selection procedure applied here? Response to Comment R2.11: We first identified 22 GCMs from the CMIP5 Coupled Model Intercomparison Project (http://cmip-pcmdi.llnl.gov/cmip5/availability.html) based on the availability and completeness of climate projections, and then included realizations of 5 GCMs (CanESM2, FGOALS-s2, IPSL-CM5A-LR, MIROC5, and MPI-ESM-LR), which led us to have 35 GCMs in this study. Adding more details, when we obtained the GCM outputs in 2013, some GCM outputs were not available as they were under investigation and use for their projects. In addition, we did not include GCMs that do not provide all scenarios of our interests, such as historical, RCP 4.5, and RCP 8.5.

Comments R2.12: P4-L16: "locations": What do you mean here? Response to Comment R2.12: What we meant by "location" was "geographic location". We will modify the word to "geographic location" in the revision.

Comments R2.13: P5-L27: what is meant with the long-term monthly hydrologic response? Response to Comment R2.13: The "long-term monthly hydrologic responses" would be represented by monthly hydrographs of direct runoff, soil water, evapotranspiration, and groundwater of a watershed over long-term periods such as 2020 to 2099.

Comments R2.14: P5-L28: WW is an often used indicator for water withdrawals. Better use WA here. Response to Comment R2.14: Following the suggestion, we will use "WA" to denote available water or water available in the revised manuscript.

Comments R2.15: P7-L11: "maximum and minimum values": please specify where these values refer to Response to Comment R2.15: For the purpose of clarification, we modified the sentence to "The difference between maximum and minimum values of projected climate variables (precipitation and temperature) and hydrologic components (QQ, DR, SS, etc.) per each month was calculated as a measure of the overall amounts

of uncertainty for the corresponding month in the ensemble predictions made using multiple GCMs and behavioral parameter sets.

Comments R2.16: P7-L23 (formula): where do the subscripts mentioned in the formula come from? Cannot find their meaning in the text. Response to Comment R2.16: The subscripts came from the set theory of mathematics. x means an element (a climate model or hydrologic model parameter), S is a set (a set of climate models or hydrologic model parameters) of elements, âĹĂx âĹĹ S signifies all elements in S, and x âĹĽ S represents that an element x is not in S. For clarification, we will add this description to the revised manuscript. (See equations in an attached pdf file.)

Comments R2.17: P8-L19: "03232500": Doesn't this watershed have a name? Response to Comment R2.17: It may have a local name, but we decided to use the identification number assigned to each watershed by USGS.

Comments R2.18: P9-L5-6: I do not understand how to interpret these values: are these the average projected increase rates across all hydrologic components? Response to Comment R2.18: The values are averages of projected increases (or changes) of each hydrologic component (e.g. QQ) across the study watersheds. The variations over the 61 watersheds are represented with box-whiskers by hydrologic components in Table 2.

Comments R2.19: P9-L7-8: "indicating. . .runoff hydrographs": What could be a reason for this observation? In most models you see that precipitation changes are buffered towards runoff estimates. Response to Comment R2.19: As described in the manuscript, the amount of precipitation was projected to increase by 6.8% on average in the study watersheds under the RCP4.5 scenario, and the increase rates were higher in winter and spring when soil water content was relatively high compared to that of summer. In addition, the temporal variations in DR (direct runoff) projections were greater than those of GW (groundwater), and the variations in QQ (total runoff) projections were dominated by those of DR. Thus, the precipitation increase could be

amplified in DR by being concentrated on seasons with high soil water contents. We will add this description right after the sentence, which reads "This amplification happens as the projected precipitation increases concentrate on winter and spring (Figures 3(a) and 3(b)) when soil water content is relatively high (Figure 6: SS) in the study watersheds".

Comments R2.20: P9-L11-13: "Implying. . .PP": Where does this water go to then? Increased ET? Higher soil moisture? Or less low-period flows? Response to Comment R2.20: The available water precipitation (PP) and soil moisture (SS), and decrease in SS will cancel out the increase in PP in the overall water balance. Specifically, as seen in Figures 7 and 8, the increase rate of PP is a bit greater than that of evapotranspiration (ET), and the difference between them becomes a part of the increase (1.5% to 1.6% depending on RCPs) in the amount of available water (WW). In addition, as seen in Figure 8, the rates and directions of changes in WW and SS to PP vary over months.

Comments R2.21: P9-L28: "multi-model(multi-GCM): just give it one name: multi-GCM. Response to Comment R2.21: As recommended, we will use the single term, "multi-GCM", in the revised manuscript.

Comments R2.22: P10-L9: I do not fully understand the idea behind the overall parameter posterior distribution. What is the added value to aggregate the results over all the watersheds? Response to Comment R2.22: As many watersheds were included in the study, it was quite difficult to include all the posterior distributions developed for individual watersheds in the limited space. However, we thought it would be still good to show how the overall posterior distributions looks like so that we could give readers an idea of how the parameter uncertainty of the ABCD models look like.

Comments R2.23: P10-L24: "Unit of depth": Not clear what is meant with this Response to Comment R2.23: What we meant by "unit of depth" is that we calculated the amount of hydrologic components in the depth unit of "mm" rather than discharge such as "cms". For clarification, we will modify the term to "in the depth unit of mm."

Comments R2.24: P11-L5: "relatively constant .. than in summer": sounds like a contradiction Response to Comment R2.24: We agree with that the expression is not clear. We will revise the sentence to "alternatively, the uncertainty of the actual ET projections was larger in winter than in summer, but the seasonal variations were small."

Comments R2.25: P11-L26: Not clear to me whether a threshold of 97.5% is relatively loose or conservative. Response to Comment R2.25: We defined 97.5% as a relatively liberal or loose threshold compared to 90.0% as it will provide a smaller estimate of uncertainty amount.

Comments R2.26: P11-L29: In the ABCD model soil moisture and groundwater are 'rest-products', isn't it straightforward then that the impact of model-parameterization is larger than the impact of uncertainty in GCM-input? Response to Comment R2.26: Not necessarily. The soil moisture and groundwater components of ABCD are treated as important as the other hydrologic components since direct runoff is expressed as a function of these components in the model. Thus, soil moisture and groundwater simulated using ABCD will have the same level of accuracy as that of the other simulated components. Of course, we may have different findings on the relative impacts if a different hydrologic model is used, but we think the results are reasonable as soil moisture and groundwater are relatively less responsive to rainfall and temperature than direct runoff due to the water holding capacity of soils.

Comments R2.27: P11-L30: How can ET be directly be determined by direct runoff? Shouldn't this be the other way around: uncertainty in QQ being driven by uncertainty in ET? Response to Comment R2.27: By a mass balance (or continuity equation), in ABCD, ET is regarded as the difference between the precipitation and the total runoff simulated with consideration the nonlinear relationship among other hydrologic components. Uncertainty in QQ is driven by those of DR and GW as Eq. 8 represents. As QQ is a function of DR, GW, and SS, ET will become dependent on DR, GW, SS as well as PP (Eqs. 2 to 8: Thomas, 1981; Sankarasubramanian and Vogel, 2002; Martinez and Gupta, 2010). References Thomas, H.: Improved methods for national

water assessment, Report WR15249270, US Water Resource Council, Washington, DC, 1981. Sankarasubramanian, A. and Vogel, R.M., 2002. Annual hydroclimatology of the United States. Water Resources Research, 38(6), WR000619.1-12. Martinez, G.F. and Gupta, H.V., 2010. Toward improved identification of hydrological models: A diagnostic evaluation of the "abcd" monthly water balance model for the conterminous United States. Water Resources Research, 46(8), W08507.1-21.

Comments R2.28: P12-L10: So would you say that, in the Ohio River basin, precipitation is a larger determinant for water availability than Temperature? Response to Comment R2.28: The results did not provide a strong evidence supporting that precipitation is a large determinant for water availability than temperature. The results said that uncertainty in the hydrologic components would be more significantly affected by uncertainty in the GCM-ensemble projections of precipitation than temperature. This occurs when the variations of precipitation projections are relatively very large compared to those of temperature projections even when hydrological processes of a watershed are mainly dominated by temperature. The amount of uncertainty in the ensemble climate projections is determined by the feature of the climate models rather than the watersheds, thus the results can be opposite if uncertainty in temperature projections is greater than that of precipitation projections. The results showed that in the study watersheds the amount of uncertainty in the precipitation projections is much greater than uncertainty in the temperature projections made using the climate models.

Comments R2.29: P12-L13-14: "This finding. . .climate change": This is an observation that hold for this basin specifically. Does it also hold for other types of catchments, e.g. the temperature dominated ones? Response to Comment R2.29: As mentioned in Response to Comment R2.28, it must depend on the amount of uncertainty in climate projections and the physical characteristics of a study watershed. The uncertainty of precipitation projection can be a major source of uncertainty in hydrologic component projections even in a temperature dominated watershed when the amount of precipitation projection uncertainty is much greater than that of temperature projection uncertainty.

Comments R2.30: P12-L19-20: "and the thresholds. . .climate change": Incomplete sentence Response to Comment R2.30: The sentence will be revised for clarification by adding "on" between "interest and" and "the thresholds".

Comments R2.31: P13-L1-3: "A total of. . . explored": Fuzzy sentence, please rephrase. Response to Comment R2.31: We will revise the sentence to "A total of 22 GCMs and their variants were considered in this study so that wide ranges of mathematical representations and the simulation strategies of climate processes could be considered, and the largest uncertainty in the multi-GCM ensembles could be explored".

Comments R2.32: P13-L3-4: "Uncertainty associated. . .amount of precipitation": I don't understand this statement. Please clarify. Response to Comment R2.32: We will modify the sentence to "When the amount of uncertainty is expressed in the depth unit (mm), it was turned out that uncertainty in hydrologic component projections made using the multi-GCMs is considerably large compared to the rainfall depth projected at monthly and annual scales, indicating a GCM selection can substantially affect a hydrologic analysis of climate change", which is demonstrated in Figure 11.

Comments R2.33: P13-L7: How about the regional scale climate projections? Response to Comment R2.33: This study focused watersheds in the average size of 678 km2 (standard deviation of 817 km2) ranging from 47 to 6,037 km2. The hydrologic responses of large watersheds or basins tend to be dominated by slow hydrologic components such as groundwater and soil water rather than direct runoff. The results of this study demonstrated that groundwater and soil water components are less sensitive to uncertainty in the GCM ensemble. Thus, the impacts of uncertainty in ensemble precipitation and temperature projections on hydrologic analysis of climate change are expected to be smaller at regional-scale than at local-scale. We will add this discussion in the revised manuscript.

Comments R2.34: Fig1: Is it necessary to present the area size in terms of log10? Response to Comment R2.34: Yes, our trials showed that the log-scale plot can better depict the distributions of the watershed areas as the areas exponentially distributed.

Comments R2.35: Fig5: These sub-figures are very difficult to compare. Would it be possible to make 2 figures (one for precip and one for temp) showing the ltm-hydrographs (12 months) for the different scenarios: * current conditions * near future under rcp4.5 and rcp8.5 * far future under rcp4.5 and rcp8.5 Response to Comment R2.35: We understand your concern. However, our trials showed that combining the eight plots into two made plots look busier and difficult to read lines and symbols. Then, we decided to keep the current forms even though they take up space.

Comments R2.36: Fig 6: see comment figure 5. Response to Comment R2.36: We got the same conclusion as that of Figure 5. Instead, we decided to remove a half of them (for RCP 4.5) so as to improve the readability of the plots.

Comments R2.37: Fig 7: Does it really make sense to show a multi-GCM, multi-parameter, multiwatershed projection? Response to Comment R2.37: We believe Fig. 7 helps us to understand the seasonal variations in the projected changes of the study watersheds' hydrologic components, providing the overall summary of the projections. Thus, we think it is important to include it in the manuscript.

Comments R2.38: Fig 9: Are these values for all watersheds? Response to Comment R2.38: Yes, as mentioned earlier, it was not efficient to include all the posterior distributions developed for individual watersheds in the limited space. Thus, we decided to show how the overall posterior distributions look to give readers an idea about the parameter uncertainty of the ABCD models used in this study.

Comments R2.39: Fig 12: Could you explain why for GW and PET the sign changes when the threshold increases? In lower thresholds multi-parameter uncertainty shows to be more important, whilst in high thresholds multi-GCM is a more important determinant for the uncertainty in outcomes. Response to Comment R2.39: For the purpose

of clarification, we want to clarify that the sign does not change across the thresholds, but the ratios drop below one. The understanding of the reviewer is correct. While a threshold increases from 90.0% (conservative) to 97.5% (liberal), the relative contributions of the multi-GCM ensemble to the overall uncertainty increase as the estimated amount of uncertainty in hydrologic modeling decrease.

Comments R2.40: Fig 13: Legend of this figure is difficult to interpret. Perhaps give a figurative example to clarify. Response to Comment R2.40: Following the suggestion of the reviewer, we will add a figurative example to improve the readability of the figure.

Please also note the supplement to this comment:
http://www.hydrol-earth-syst-sci-discuss.net/hess-2016-160/hess-2016-160-AC2-supplement.pdf

———————————————